# Assessing the Factor Structure of the Farm/Ranch Stress Inventory in a Sample of LGBTQ+ Farmers Across the United States

**DOI:** 10.3390/ijerph23010022

**Published:** 2025-12-23

**Authors:** Matthew Rivas-Koehl, Courtney Cuthbertson, Dane Rivas-Koehl, Anisa Codamon, Asa Billington

**Affiliations:** 1Department of Human Development & Family Studies, University of Illinois, Urbana, IL 61801, USA; mr36@illinois.edu (M.R.-K.); mdr8@illinois.edu (D.R.-K.); acodam2@illinois.edu (A.C.); 2Department of Counseling, Human Development and Family Science, University of Vermont, Burlington, VT 05405, USA; asa.billington@uvm.edu

**Keywords:** farm stress, mental health, LGBTQ+, measurement, principal components analysis

## Abstract

Farmers face disproportionately high levels of stress, which has implications for their mental health and well-being. Notably, existing measures of farm stress have not attended to variations in stress based on identity-related experiences. This study investigated the factor structure of the Farm/Ranch Stress Inventory (FRSI) among LGBTQ+ farmers in the United States to see whether the factors were consistent with previous studies. We surveyed the experiences of 148 LGBTQ+ farmers from across the U.S. and used principal components analysis to analyze the factor structure of the FRSI. A scree plot, parallel analysis, variance explained, and interpretability factors were considered to determine the best-fitting solution. Our analysis revealed a six-component structure: Family and Finances, Operational Challenges and Sustainability, Long-Term Farming Viability, Social and Safety Dynamics, Logistical Circumstances, and Cultural Isolation, which diverged from the original five-factor model of Finances, Government and External Stress, Work Stress, Operation Stress, and Isolation. This study highlights that measures of stress must consider the unique contexts that may shape the experiences of farm stress among diverse populations. The results suggest that future measures of farm stress used with LGBTQ+ farmers might consider cultural and social factors in particular to better understand their experiences of stress.

## 1. Introduction

Understanding the unique challenges and stressors that farmers face is imperative. Previous research has often attempted to understand farm stress without explicitly considering contextual factors, such as sociocultural conditions and sociodemographic characteristics. In particular, there is a dearth of literature about the experiences of LGBTQ+ farmers, and emerging evidence suggests this population may struggle disproportionately with stressors that can adversely affect mental health (e.g., [1]). Existing measures of farm stress have not been evaluated for use with LGBTQ+ farming populations who may face stressors unique to their marginalized identities. To fill this gap in the literature, we examined the factor structure of the Farm/Ranch Stress Inventory (FRSI; [2,3]) in a sample of LGBTQ+ farmers to determine whether the results were consistent with testing of the original measure.

### 1.1. Farm and Ranch Stress

Farm stress is an important consideration affecting the health and well-being of farmers [4]. The stress experienced by farmers is thought to contribute to a suicide rate that is among the highest of any subpopulation in the United States (U.S.; [5]). Farmers face stressors such as unforeseen weather, physical health issues, finances, and time pressure that contribute to mental health concerns, including depression and anxiety [6,7,8]. In the agricultural industry, those involved in farming are expected to act independently, be tough, and not ask for help, as this can be interpreted by other farmers as a sign of mental and physical weakness [9,10,11,12]. Such norms present in agriculture have led to a stigmatization of discussions related to mental health. Programs aimed to support farmers’ mental health and well-being exist [13,14,15], and to bolster these programs, continued research that understands the unique stressors they face is needed.

### 1.2. LGBTQ+ Farmers

Studies have repeatedly documented that lesbian, gay, bisexual, transgender, and/or queer (LGBTQ+) people experience worse mental health than cisgender, heterosexual peers [16,17], similar to the disproportionate rates of adverse mental health among agricultural producers compared to people in non-agricultural occupations [5,18]. Hegemonic narratives about U.S. agriculture are rooted in white, patriarchal, heteronormative family systems [19] and often leave out LGBTQ+ people. How, where, and with whom LGBTQ+ people farm is shaped by their LGBTQ+ identities, including considerations such as land access, financial capital, and learning farming practices [20]. For some, agriculture has enabled LGBTQ+ people to create intentional communities [21] and to break away from gender and sexual norms [22]. Networks of LGBTQ+ farmers are a source of information, social connection, and solidarity [23]. For example, LGBTQ+ farmers report navigating heterosexism by selectively coming out to others and building formal and informal queer farmer networks.

Still, however, heteronormativity, heterosexism, and homophobia negatively impact LGBTQ+ people who want to and do farm [22]. Many agricultural resources are misaligned with the identities or practices of LGBTQ+ farmers [23] or are less readily available to them [19]. Likely resulting from these stressors and lack of resources, recent research has found LGBTQ+ farmers may experience dual risks for adverse mental health outcomes [1]. Specifically, this research found scores for diagnoses of depression (36%) and anxiety disorder (46%) at rates far exceeding both general farmers’ and LGBTQ+ community estimates. The same study found over half (52%) of the sample to be at risk for suicide. 

### 1.3. Measuring Farm Stress

Despite the consensus that agriculture is a stressful industry, researchers have not used one standardized measure to assess various sources of occupational stress among farmers. One of the earliest measures of farm stress was the Farm Stress Inventory (FSI; [24]), based on an interview study with 140 Canadian farmers and their spouses. The FSI included 60 items, where participants responded on a five-point scale about how frequently they experienced stress, ranging from “almost never” to “almost always.” The 60 items could be summed to make one farm stress score, which significantly correlated with depression symptoms [25].

Additional farm stress measures were constructed and validated throughout the 1990s and into the early 2000s. Eberhardt and Pooyan [26] developed the 28-item Farm Stress Survey (FSS) from unstructured interviews with farmers. The FSS asks participants to rate how much each item was a source of concern or worry on a seven-point scale, ranging from “not at all” to “an overwhelming extent.” The FSS is reduced to six factors: personal finances, general economic conditions, geographic isolation, time pressure, climatic conditions, and hazardous working conditions. The FSS was validated using a sample of 362 midwestern U.S. farmers; the sample was predominantly male, with an average age of 50 years and a high school level of education.

Araquistain [27] developed the Farm/Ranch Stress Scale (FRSS), a 13-item scale that asks participants to report on the perceived stress their family experiences in agriculture. For each item, participants provide a rating on a four-point scale from “no stress” to “very stressful.” All items are summed to create a scale where higher scores are indicative of greater perceived stress. The FRSS was developed using a sample of 188 men and women from farm families, including 86 couples (men and women who were married to each other), from nine counties in Idaho. Ide et al. [28] found the FRSS was correlated with family strain, discord, and quality of life, but suggested a revised 22-item scale based on analysis of open-response questions. The revised FRSS included items like “inheritance problems,” “time management and schedules (deadlines, lack of schedule, not enough time, no free time, sleeping problems, balancing outside work and ranch demands),” and “agricultural economy (prices, getting and dealing with buyers, lack of stability, future)” [28]. The revised FRSS was validated using two samples of heterosexual, married farm couples from Idaho.

The Edinburgh Farming Stress Inventory (EFSI; [29]) was created and validated on a sample of 318 farmers in the U.K. The measure was adapted in part from Eberhardt and Pooyan [26] and included 10 items related to farming stress, specifically addressing government policy, time pressure, hazards in farming, unpredictable factors, finances, and geographical isolation. Participants were asked to rate the severity of their stress from each of the sources, and they responded to each item on a five-point scale, ranging from “none” to “very severe.” The overall measure is divided into six factors, including farming bureaucracy, finance, isolation, acts of God, personal hazards, and time pressure. Deary et al. [29] found age to be inversely related to most farm stressors.

Welke [2] posited that existing measures had been sparsely used, not validated, or had limitations; Welke then developed the Farm/Ranch Stress Inventory (FRSI) by combining some items from both the FSS and FRSS. The FRSI included 27 items about different aspects of farming or agriculture and asked participants to respond about how much stress it caused their family, with responses on a four-point scale from “no stress” to “very stressful.” Three additional open-response questions were included at the end of the inventory for participants to write in a stressor not listed previously. The first version of the FRSI [2] was validated in a sample of 321 South Dakota farmers or ranchers; while the sample design was intended as a dyadic survey of the primary operator and their spouse, 242 surveys were returned together and 231 participants reported being married to each other. Of dyadic survey packets returned, only two were from pairs of same-sex participants and in both instances, one person within the dyad indicated not being married. The average age of the sample was 51.17, and 96% were white. With this first sample, the 27 items loaded onto five factors: finances, government and external stress, work stressors, operation stressors, and isolation.

In 2004, Welke [3] updated the Farm/Ranch Stress Inventory to include a 28th item about healthcare costs and modified the instructions to ask participants to respond regarding how much stress each item caused them (rather than their family). The updated inventory was validated on a sample of 265 South Dakota farmers/ranchers, and used a similar dyadic design. Of the 265 participants, 184 were male/female couples married to each other. The average age of the sample was around 53, and about 95% of the sample was white. Using factor analysis, Welke found the same five factors as in the previous study. A modified version of the FRSI was created by Kearney et al. [30] by correcting six instances of erroneous capitalization or grammar from Welke’s 2004 FRSI measure. Kearney et al.’s modified version included the same 28 items, on which participants responded on a four-point scale indicating the level of stress associated with each item, ranging from “no stress” to “very stressful.” One open-response question (rather than three) asked participants to list any other items they find stressful related to farming. The items were grouped by farm-related factors, financial factors, and social factors, although Kearney et al. [30] did not conduct a factor analysis to determine or test factor structure. The sample included 128 farmers from North Carolina, three-quarters of whom were male, with the largest proportion between the ages of 40–59 and with high school education or less; the sample was also nearly entirely non-Hispanic white. Modified FRSI factors have explained approximately 20% of the variance in farmers’ psychological well-being in one study [31].

Across the multiple farm stress measures created by different scholars, items have focused on specific occupational tasks, occupational characteristics, or external macroeconomic, political, or weather-related factors. Such measures have largely left out items that consider how occupational experiences are shaped by systems of power that marginalize people based on identities they hold, like sexual and gender identities. Sexual and gender minority people experience stress from navigating hostile and potentially hostile social environments [32,33,34], including in occupational settings [35]. That existing farm stress measures do not include identity-based variation in occupational stress demonstrates an assumption that occupational stress can be understood without regard for identity-related stress, and that individuals can or would want to separate their experiences of occupational stress from identity-related stress. Assessing the factor structure of the FRSI in a sample of LGBTQ+ farmers can aid understanding of whether identity-evasive measures are appropriate for analyzing occupational stress in non-majority populations.

### 1.4. The Current Study

Since many recent studies have drawn on the 2004 FRSI [3] to measure farm stress, we employed Principal Components Analysis (PCA) in SPSS Version 29 to examine the factor structure of the updated FRSI with a sample of LGBTQ+ farmers. Following calls for more research to understand how LGBTQ+ identity impacts the well-being of people in agriculture [36], the current study contributes findings at the intersection of gender, sexual identity, and farming identity by examining the measurement of farming-related stress among LGBTQ+ farmers.

## 2. Materials and Methods

### 2.1. Participants and Procedure

We recruited participants via social media and direct contact with national and regional agriculture- and LGBTQ+-related organizations who assisted in disseminating promotional materials. Agricultural organizations included the National Association of State Departments of Agriculture, National Farmers Union, the Cultivating Change Foundation, American Farm Bureau Federation, and queer farmers group. LGBTQ+-related organizations included community centers identified using maps from CenterLink, an organization for LGBTQ+ centers. Inclusion criteria were that participants must be age 18 or older, reside in the United States, self-identify as a farmer or working in agriculture, and self-identify as LGBTQ+.

Participants completed an online survey administered via Qualtrics about their identities and experiences with mental health, farm stress, minority stress, and social support. Participants were 148 LGBTQ+ farmers and/or ranchers ages 19 to 66 (*M*_age_ = 32.5, *SD* = 8.79) from across the U.S. Information about the sample has been published elsewhere [1]. The survey allowed participants to choose all sexual, gender, and racial identities that applied to them. For sexual identity, 86 (58.1%) participants identified as queer, 36 (24.3%) as gay, 33 (22.3%) as bisexual, 35 (23.6%) as lesbian, and 28 (18.9%) as pansexual. Some participants selected identities such as same-gender loving, fluid, asexual, and aromantic; the *n* for these participants was excluded to protect confidentiality (*n* < 5). The majority of participants were white (*n* = 86, 86.5%) and by gender, the largest proportion were women (*n* = 58, 39.2%). Over a third identified as transgender (*n* = 56, 37.8%). The average participant had been farming for about 7.16 years (*SD* = 7.1). Farm workers accounted for 43.9% of the sample (*n* = 65), and 33.8% of the sample were the primary owner/operators of a farm (*n* = 50). The most commonly farmed commodity reported by participants was vegetables and/or fruit (64.9%).

### 2.2. Measures

#### Farm and Ranch Stress Inventory (FRSI)

Participants responded to 28 items following the stem “Please rate each item according to how much stress it causes you…” Example stressors included “Distance from shopping centers/school/recreation, etc.,” “Farm/ranch accidents and injuries,” and “Concern over the future of the farm/ranch.” Response options ranged from 1 (no stress) to 4 (very stressful). The 28 items of the FRSI, as performed by Welke [3], have traditionally been distilled into five factors: finances, government and external stress, work stressors, operation stressors, and isolation. All items are shown in Table 1.

## 3. Results

### 3.1. Descriptive Analyses

The Kaiser-Meyer-Olkin (KMO) factor adequacy test suggested that the sampling adequacy of the data was appropriate for use with PCA (KMO = 0.82). A Scree Plot (see Appendix A), Horn’s Parallel Analysis, and Kaiser’s Rule were used to help determine the number of factors that should be retained when analyzing the PCA. The Scree Plot suggested that 7 components be retained; Horn’s Parallel Analysis suggested that 5 components be retained; and Kaiser’s Rule suggested that 8 components be retained.

### 3.2. Principal Components Analysis

We first conducted an unrestricted PCA using ProMax rotation to allow components to correlate. This resulted in an 8-component solution that the team determined inadequately represented the data given that the items were split across too many components for substantive interpretations. Given the results from the preliminary analyses, we investigated the 5, 6, and 7-component analyses further. Although solutions with more components explain more of the variance in the data, the substantive interpretation of the components was considered in tandem. Communalities were examined to detect any problematic items (i.e., items that shared especially low variance with other items); all values were satisfactory (>0.4). We determined that a six-component solution was the best fit to the data, based on our descriptive analyses, the amount of variance explained, and our ability to meaningfully interpret how items contributed to each of the components [37]. The six-component solution explained 59.1% of the cumulative variance, meaning the six components together accounted for just over half of the total variability in participants’ responses. This suggests that these components accounted for most of the patterns present in the data. The first component concerned family and financial matters; the second component included items pertaining to operational challenges of farm work, including farm sustainability; the third component was related to long-term farming viability; the fourth component concerned social and safety factors; the fifth component was related to farm logistics; and the sixth component pertained to cultural isolation. Results of the extraction are presented in Table 1. Component correlations are presented in Table 2. See Appendix A for a comparison of how items in the FRSI loaded onto factors in the original analysis [3] versus our analysis.

**Table 1 ijerph-23-00022-t001:** Factor Loadings Extracted from a PCA of the FRSI.

FRSI Item	Factor Loadings					
	1	2	3	4	5	6
Not enough cash/capital for unexpected problems (illnesses, health care, breakdowns, other emergencies)	**0.847**	0.094	−0.052	−0.093	0.089	0.002
Financing for retirement	**0.774**	−0.056	0.085	−0.280	0.111	0.310
Health care costs (direct costs and/or cost of insurance)	**0.703**	−0.057	−0.015	−0.011	0.260	0.269
Not enough money for day-to-day expenses (purchases, repairs, parts, fence and building maintenance, etc.)	**0.628**	0.271	0.078	−0.188	−0.114	−0.242
Balancing the many roles I perform as a family member and a farmer/rancher	**0.470**	0.189	−0.217	0.373	−0.074	0.115
Taxes (high taxes, figuring taxes, etc.)	**0.461**	0.107	0.432	0.035	−0.044	0.002
Market prices for your crops/livestock	**0.438**	−0.010	0.191	0.201	−0.231	−0.023
High debt load	**0.437**	0.007	0.402	0.034	−0.082	−0.251
Not enough time to spend together as a family in recreation	**0.411**	0.240	0.115	0.096	0.055	−0.127
Not having the people power to operate the farm/ranch	−0.028	**0.781**	−0.185	0.038	−0.017	0.098
Having too much work for one person	0.175	**0.748**	−0.253	−0.003	−0.036	0.107
Concern over the future of the farm/ranch	0.268	**0.609**	−0.163	0.046	0.076	−0.059
Problems with machinery (purchases, repairs, breakdowns)	0.029	**0.594**	0.326	−0.036	0.036	−0.046
Problems with livestock or crops (illness, disease, noxious weeds, rodents)	0.190	**0.453**	0.241	−0.085	−0.124	0.167
Government farm price supports	0.414	−0.343	**0.725**	−0.062	−0.124	0.177
Operating hazardous machinery	−0.210	0.119	**0.713**	0.114	−0.099	0.095
Government export policy	0.095	−0.225	**0.704**	0.198	0.087	−0.042
Working with bankers and loan officers	0.333	0.046	**0.533**	0.118	0.091	−0.204
Limited social interaction opportunities	−0.025	0.011	−0.020	**0.771**	−0.013	0.086
Lack of close neighbors	−0.144	−0.287	0.426	**0.633**	−0.006	0.097
Farm/ranch accidents and injuries	−0.125	0.128	0.219	**0.537**	−0.103	−0.185
Working with extended family members in the farm/ranch operation (parents, in-laws, children)	−0.080	0.083	0.233	**0.430**	0.089	0.278
Seasonal variations in workload (planting season, harvest, calving time, marketing time, etc.)	−0.181	0.160	0.279	−0.001	**−0.736**	0.079
The weather (inadequate/or too much rainfall, snow, hail, etc.)	0.283	−0.083	−0.306	0.355	**−0.647**	0.105
Distance from doctors or hospitals	0.137	0.048	0.019	0.410	**0.639**	0.016
Distance from shopping centers/schools/recreation, etc.	−0.024	0.168	0.155	0.423	**0.496**	−0.028
Outsiders not understanding the nature of farming/ranching	0.099	0.125	0.008	0.106	−0.101	**0.809**
Dealing with non-relative help (incompetent help, finding good help, supervising help)	−0.130	0.540	0.328	−0.047	−0.087	**0.496**

Note. Component 1 = Family and Finances; Component 2 = Operational Challenges and Sustainability; Component 3 = Long-Term Farming Viability; Component 4 = Social and Safety Dynamics; Component 5 = Logistical Circumstances; Component 6 = Cultural Isolation. Bold indicates onto which factor the item loaded.

**Table 2 ijerph-23-00022-t002:** Correlations of the Extracted FRSI Components.

Component	1	2	3	4	5	6
1	1.000	-	-	-	-	-
2	0.402	1.000	-	-	-	-
3	0.253	0.282	1.000	-	-	-
4	0.389	0.275	0.334	1.000	-	-
5	−0.703	−0.119	0.234	0.101	1.000	-
6	−0.035	0.063	0.025	0.001	0.203	1.000

## 4. Discussion

In this study, we aimed to investigate the factor structure of the Farm/Ranch Stress Inventory using a sample of LGBTQ+ farmers. There is a dearth of literature regarding LGBTQ+ people in agriculture, and to better understand this population, we must first be confident in our current measures of agriculture-related concepts. Despite previous analyses of the FRSI, it is essential to determine whether the same factors emerge within the measure among LGBTQ+ farmers. Notably, the factors that emerged in our data analysis differ from those in the original scale [3]. This could be due to methodological differences; however, the current analysis addresses previous limitations (e.g., using a rotation that allows correlated factors). Factor differences may have arisen due to contextual differences within the sample. The original FRSI was developed using a sample of “traditional” farmers; that is, mostly white, heterosexual men. Our study used a widely diverse sample of LGBTQ+ individuals who worked on farms and ranches across the U.S., including the fact that the average age of our sample was younger than that of the average American farmer [38]. It is reasonable to assert that differences in their lives could contribute to different interpretations of stress related to farming and ranching. Past studies have typically drawn from samples of farm owners/operators whereas the current study included a proportion of farm workers, which may explain some of the differences in our findings [39]. The six components we found represent distinct but related potential domains of stress for LGBTQ+ farmers. *Family and Finances* encompassed items that were related to both family and financial concerns. We suspect these items may have loaded together, given the integral nature of financial concerns to family life (e.g., [40]). *Operational Challenges and Sustainability* encompassed items that characterized the types of issues that might jeopardize one’s ability to keep the farm running over time. We hypothesize that these items may be sources of stress for LGBTQ+ farmers, given that this population has shown a commitment to sustainable and regenerative agriculture [21,41].

*Long-Term Farming Viability* encompasses stressors that could undermine a farm’s long-term success. These stressors, compared to *Operational Challenges and Sustainability*, may be related to stressors that are perhaps outside the individual’s control, such as the implications of current policies on agricultural work and their impact on the farm. *Logistical Circumstances* contains items that may pertain to the logistics of working in agriculture. For example, stressors such as weather and seasonal variations in workload have been documented as unavoidable stressors for agriculturalists [6,7,8].

*Social and Safety Dynamics* encompassed items related to social relationships, but also included a stressor related to safety dynamics. Particularly for LGBTQ+ farmers, who face marginalization in agricultural spaces, safety may be a concern if they do not feel adequately connected to potential sources of support in case a safety issue were to arise. Prior research highlights a potentially related phenomenon wherein rural LGBTQ+ populations face health disparities that may be related to the intersection of their social identities and the limited resources available in some rural areas [42]. Importantly, however, rural LGBTQ+ populations are not a monolith, and not all LGBTQ+ farmers live and work in rural environments. Finally, and perhaps most unique to this population, we found a component known as *Cultural Isolation*. Although represented by just two items, we posit that these may be important in characterizing farm and ranch stress for LGBTQ+ people in particular.

Both *Social and Safety Dynamics* and *Cultural Isolation*, as constructs more broadly, may relate to the fact that farmers report navigating their interactions with the agricultural industry cautiously and intentionally, considering their identities within the broader agricultural industry in ways unique to LGBTQ+ farmers compared to the general population of farmers [19,23]. Within the LGBTQ+ community, agriculture may not be perceived as a normative choice of occupation, potentially isolating individuals from other LGBTQ+ communities. Relatedly, Halberstam [43] coined the term “metronormativity” to describe the assumption that the lives of LGBTQ+ people are centered in urban environments. Although not inherently related to rural versus urban living, a similar bias toward what is normative for LGBTQ+ people to choose for their occupations may underscore the meaningfulness of the *Cultural Isolation* component found in the present study. In addition to facing norms regarding living as an LGBTQ+ person, within agriculture, LGBTQ+ individuals may not feel affirmed and may even actively experience discrimination, thus making it harder to find help (for owners) or find work (for non-owners) with others who are affirming. The contextual understanding of these stressors is important for gaining a holistic view of the stress experienced by LGBTQ+ farmers when considering both their LGBTQ+ and farming identities and follows guidance from scholars who argue for work with LGBTQ+ populations to consider all the identities that LGBTQ+ individuals hold [17]. 

Our study was limited by a relatively small sample that was not nationally representative; thus, we used PCA rather than exploratory and confirmatory factor analyses. Additionally, the current study focused solely on quantitative data and did not include participants’ responses to open-ended questions about the additional stressors they experienced. Still, the contribution of this study is noteworthy given its novelty. Future work about farm stress could improve on existing research by incorporating more measures related to social aspects of agricultural work. LGBTQ+ farmers’ experiences are shaped by navigating social relationships and processes required to enable agricultural work. Past research has noted the relational nature of agriculture and challenges LGBTQ+ farmers face due to heteronormativity in access to land and financial capital [39]. Researchers could create items drawing from minority stress theory [32] that posits sexual minority people experience additional stress from navigating hostile or potentially hostile environments.

## 5. Conclusions

Supporting agricultural efforts requires understanding occupational stressors, with attention to contextual factors such as the identities that shape individual experiences. The current study has implications for research and practice. First, our study contributes to the research literature by advocating for the examination of established measures when applied to underacknowledged populations. Second, the factors that emerged may represent areas in the lives of LGBTQ+ farmers that require interventions, ultimately supporting the health and well-being of LGBTQ+ farmers. We encourage researchers to continue to assess measures after they have been developed and established to ensure that measures accurately reflect the lived experiences of participants. We suggest that future measures of farm stress include attempts to identify unique stressors faced individuals with diverse identities, challenging the white, cisheteropatriarchal norms of the agricultural profession.

## Data Availability

To protect confidentiality of participants, data are not publicly available.

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
