# Peer review of "Assessing the Factor Structure of the Farm/Ranch Stress Inventory in a Sample of LGBTQ+ Farmers Across the United States"

_ijerph, 2025, doi:10.3390/ijerph23010022_

Round 1
Reviewer 1 Report
Comments and Suggestions for Authors
This paper examines the applicability of the Farm/Ranch Stress Inventory to an important, understudied farming subpopulation: LGBTQ+ farmers. Specifically the authors examine whether similar factor structures are present for LGBTQ+ farmers compared to the original inventory assessed on a general farming population. Their analysis found a six-component factor structure, highlighting an additional cultural isolation factor not found in the original five-factor model. This finding is notable in highlighting how marginalized farmers will experience unique contexts relating to minority stress, and is important to consider in future study of farmer stress to improve measurement among marginalized farmer subpopulations (and from a public health perspective, key to understanding stress mechanisms of farming populations overall).
I believe this is an important topic, and I enjoyed reading this work. Below I offer some suggestions and areas where clarification would add to the strength of the manuscript:
Introduction
Lines 39-41: The end of the introduction paragraph needs a bit more connection to set up the importance of the paper. Why does the FRSI matter? (It does, just highlight why to the reader). “Current measurement of farming stress, such as the FRSI, does not account for structural inequalities and social conditions faced by farmers from marginalized groups…Thus, we examined...” (or something similar).
Lines 67-69: “Many agricultural resources are misaligned with the identities or practices of LGBTQ+ farmers [22] or are less readily available to them [18].” If there’s enough room in the word count, can you expand to include how, with very brief examples from the cited research (Hoffelmeyer here or others cited elsewhere)? It would give readers context and add impact with examples of these disparities and constraints faced by LGBTQ+ farmers.
1.3 Measuring farm stress: Is there little agreement on measures OR have different researchers measured farm stress differently (or developed measures based on one another’s scales) across time? If the former, cite something or explain evidence that supports that argument (that you set up here). If the latter, re-word your topic sentence and make any relevant connections clear throughout this section—you have some of these connections explained already. Either way, I think this section could be strengthened with a clear narrative focus to guide the reader either in explaining 1) how there has been little agreement in farm stress measurement across time; or 2) how farm stress measurement developed across time and your motivation for further improving it.
Also throughout this subsection: depending on how you address the above, I would advocate for including a brief sociodemographic profile for the farming population that each measure was validated against. You do this for some, but not all. If you do it for all, or instead summarily discuss how the demographic profile of these samples is similar, I believe it could serve/emphasize the motivation for your study. Include the study location/country for each study here (you have with most but not all).
Around line 82-83: Depending on above, I suggest a transition or topic sentence to set up discussion/context of these multiple different farm stress measures. “Additional farm stress scales have been developed and validated over time…” (etc). Something to scaffold a narrative flow to reduce the reading of one study explained per paragraph.
Around lines 95-97: What are some examples of topics included in the revised 22-item scale from Ide et al.? Just a few relevant examples to show the reader context of how this expanded/changed from other measures, which would seem part of why you mention it.
Around line 107: Since the FRSI is the focus of this paper, can you give/cite the motivation/development process for the original creation of this scale (briefly, of course)? Was it based on any of the previously mentioned scales? Make these connections for the reader.
Around line 115: Can you clarify why does Kearney’s modified FRSI contain the same number of items as the original FRSI if they adapted additional measures from Eberhardt and Pooyan?
Line 120: Replace run with conduct. I would do this throughout the manuscript.
Line 131: I think this speaks a bit beyond what this specific study accomplishes, only in that there are not explicit measures of how LGBTQ identity impacts farmers, no discussion or measures of gendered farm stress, etc. But this study contributes to scholarship at the intersection of sexual and gender identity and farm population health by examining the measurement of farming-related stress among LGBTQ+ farmers, which is still an important contribution.
Materials and Methods
Recruitment methods:
Which regional or national ag organizations? Illinois, Midwestern, others? This would influence your demographics and is good practice to detail. Can you provide examples of which LGBTQ organizations? If recruitment and survey information are detailed in an existing publication, cite that here and refer curious readers to it.
Paragraph staring on line 142: I assume this was an online survey? How was it administered? When was it administered? Can you provide any other racial or ethnic group demographics? Hispanic or Latino/x? You mention respondents were from across the US—where specifically? Did you have more respondents from a certain region or state? I know your N is small--any limitations can be re-framed as opportunities for future research that encapsulates experiences of farmers from other marginalized groups (or intersectional experiences).
Do you have information on income or farm size/type (even just a breakdown between small and large-scale)? Or perhaps there is not much variation here?
I am curious about the motivation for recruiting including self-identifying as a farmer OR working in agriculture. Can you distinguish between these two groups--was it a question that was asked? My concern would be that those who identify as the latter (“working in agriculture”) could be in a wide variety of occupations—they very well could not be farmers so some of the factors and stressors (financial, operational stress components, governmental/policy, farm sustainability, etc.) would not be present for them in the same ways.
I would expect to see the demographics, even in a brief table, in the results section under descriptive analyses. Or include a discussion of how respondent demographics differ from the farming general population and why, and how they compare to what we know about the LGBTQ+ farming subpopulation (even if limited).
Line 151: Missing a period at end of sentence
Results
Line 171-172: update to “...team determined inadequately represented…”
Line 177-178: Clarify this more.
Line 186 and in the discussion: I understand items loaded on different factors, but substantively my first thought was--what is the difference between farm sustainability (under operational challenges) vs farm viability? These are the same concept to me so I am surprised to see items load in different factors. But, given your explanations of these components later in the discussion, it may add clarity to name one of them differently. Maybe add other terms from your explanation, so it would be “Long-term Farming Viability” or “Policy and Farming Viability”? Something to better distinguish between the second and third components.
Discussion
Line 219: Update agricultural to agriculture
I was also curious on how, substantively, to distinguish between Social and Safety Dynamics and Cultural Isolation, since some very similar items loaded on different factors, so I am glad to see this discussed at length. The discussion in this part could be bolstered by including insights from previous research on rural LGBTQ people (although not exclusive to farming populations but still relevant to social/cultural/geographic isolation or social support). Woodell 2018 is an overview of rural sexual minority health disparities and population health (and social support) that may be a helpful starting point for relevant citations to draw on.
In addition, to cite evidence to support lines 238-239 you could include something on the urban bias in the LGBTQ community, relating it to your discussion of isolation if pursuing an ag occupation. Perhaps Halberstam’s queer metronormativity or something on queer anti-urbanism.
At the end, I think you could close by re-emphasizing the argument that you had set up in the beginning-- that measurements of farming stress should encapsulate marginalized experiences, such as access to social/cultural and material support, or feeling like one belongs in the farming community. Since current farming stress measures were developed with the “default” experiences of a white, cisheteropatriarchal farming community in mind, this misses other sources of farm stress for those on the margins. Measures should and could be included on general farm stress scales to identify sources of farm stress among all members of the farming community. I think this aligns with your argument in the beginning, but if this is a digression from your aims, please disregard.
Author Response
Thank you very much for your thoughtful feedback on our manuscript. We have compiled all comments and our responses into one document, which is attached. We are positive the manuscript is stronger because of your feedback.

Reviewer 2 Report
Comments and Suggestions for Authors
This topic may be of interest to readers and is a rigorously analyzed dataset. The importance is in presenting a subset of the population that has not been studied.
There are items to clarify that would make it stronger:
- It would be helpful to know the farming experience of this group. Do they own farms, work on others' farms, woof? Did they grow up on farms? How similar or dissimilar are they to the other groups who have been studied? and how representative are they of LGBTQ populations? Situating this sample in other similar studies, as regards demographics and work experience, is important. Otherwise the meaning of these results is hard to interpret.
- This population is very young - much younger that other ag populations who have been studied. This should be noted as a significant limitation of the work--one wonders if the differences are due to age and not to gender/identity issues. (similar problem to #1, above)
- Inclusion of responses to open ended questions could be of interest.
Author Response

(The authors gave the same response as above.)
